# The Association between Serum Bilirubin Levels and Colorectal Cancer Risk: Results from the Prospective Cooperative Health Research in the Region of Augsburg (KORA) Study in Germany

**DOI:** 10.3390/antiox9100908

**Published:** 2020-09-24

**Authors:** Nazlisadat Seyed Khoei, Gabriele Anton, Annette Peters, Heinz Freisling, Karl-Heinz Wagner

**Affiliations:** 1Department of Nutritional Sciences, Faculty of Life Sciences, University of Vienna, 1010 Vienna, Austria; nazlisadat.seyedkhoei@univie.ac.at; 2Institute of Epidemiology, Helmholtz Zentrum Munich, 85764 Neuherberg, Germany; gabriele.anton@helmholtz-muenchen.de (G.A.); peters@helmholtz-muenchen.de (A.P.); 3Nutritional Methodology and Biostatistics Group, Section of Nutrition and Metabolism, International Agency for Research on Cancer (IARC-WHO), 69372 Lyon, France; freislingh@iarc.fr

**Keywords:** bilirubin, unconjugated bilirubin, cancer, colorectal cancer, antioxidants, KORA

## Abstract

Emerging studies have suggested that bilirubin, particularly unconjugated bilirubin (UCB), has substantial anti-inflammatory and antioxidant properties that protect against oxidative stress-associated diseases such as cancer. Few observational studies have investigated the etiological role of bilirubin in colorectal cancer (CRC) development. In this case-control study, nested in the population-based prospective cohort of the Cooperative Health Research in the Region of Augsburg (KORA) study in south Germany, pre-diagnostic circulating UCB concentrations were measured by high-performance liquid chromatography in 77 CRC cases and their individually matched controls. Multivariable unconditional logistic regression was used to estimate the odds ratios (OR) and 95% confidence intervals (CI) for associations between log-transformed UCB levels (log-UCB), standardized per one-standard-deviation (one-SD) increment, and CRC risk. The models were a priori stratified by sex based on previous evidence. In the fully adjusted models, each one-SD increment in log-UCB was indicative of a positive association with CRC risk (OR, 1.20; 95% CI, 0.52–2.79) among men, and of an inverse association (OR, 0.76; 95% CI, 0.34–1.84) among women (P_heterogeneity_ = 0.4 for differences between men and women). We found little evidence for sex-specific associations of circulating bilirubin with CRC risk, and further studies are needed to confirm or refute the potential associations.

## 1. Introduction

Colorectal cancer (CRC) is one of the leading causes of cancer mortality worldwide. There were over 1.8 million new cases in 2018, and CRC rates are substantially higher in men than women [1].

Oxidative stress, expressed as a constant increased reactive oxygen species (ROS) load, and inflammation are involved in the development of a variety of diseases including cancer [2]. CRC is tightly associated with chronic inflammation, and inflammatory cells can also trigger the production of ROS, which can increase the risk of mutagenesis for nearby cancer cells [3,4]. CRC might thus be a candidate for prevention by anti-inflammatory and antioxidant agents.

Emerging studies have suggested that endogenous bile pigments such as bilirubin, particularly unconjugated bilirubin (UCB), have substantial anti-inflammatory and antioxidant properties that protect against oxidative stress-associated diseases such as CRC [5,6,7,8,9]. Results from our previous in vitro studies also showed that UCB has anti-mutagenic properties [10], which may be particularly relevant for bowel health. Intestinally abundant tetrapyrroles, which are part of the bile pigment family, prevented the genotoxicity induced by poly-/heterocyclic amines and induced apoptosis in cancer cells [11,12,13,14].

Serum bilirubin is derived primarily from the degradation of hemoglobin and transported to the liver by binding to albumin. Within the hepatocytes, uridine diphosphate glucuronosyltransferase 1A1 (UGT1A1) is an enzyme that contributes to bilirubin glucuronidation [15]. Bilirubin glucuronides undergo deconjugation in the bowel by bacterial and mucosal glucuronidases for reabsorption or excretion [16,17]. Glucuronidation is essential for the biliary elimination of bilirubin, and a polymorphism in the *UGT1A1* gene results in the accumulation of UCB in the serum. Mild unconjugated hyperbilirubinemia, known as “Gilbert’s syndrome”, has a prevalence of 5–10% in Caucasians and is associated with a polymorphism of the 5′ end of the *UGT1A1* gene promoter, a homozygous insertion of thymine–adenine (TA) pairs (genotype *UGT1A1**28/*28) [18]. Findings from epidemiological studies regarding the association of circulating bilirubin levels and CRC risk remain largely controversial [19,20,21,22,23,24,25]. This may be partly attributable to the fact that the studies considered total bilirubin and were cross-sectional or retrospective in design, with two exceptions [21,25]. We recently reported on associations between UCB and CRC risk in the European Prospective Investigation into Cancer and nutrition (EPIC) study and found that higher circulating UCB concentrations, as measured in our lab, were positively associated with CRC risk in men and inversely associated with risk in women [25]. Here, we aim to replicate our previous analyses using a similar design but with data from an independent cohort with no overlap of participants and with UCB measured in the same lab. We analyzed pre-diagnostic circulating levels of UCB in relation to CRC risk in a case-control study, nested within the prospective Cooperative Health Research in the Region of Augsburg (KORA) cohort in south Germany.

## 2. Material and Methods

### 2.1. Study Population and Collection of Blood Samples and Data

KORA is a regional research platform for population-based surveys. Between 1985 and 2001, four cross-sectional health surveys (S1 to S4) were performed, each comprising an independent random sample of all inhabitants of German nationality aged 25–74 years from the region of Augsburg, South Germany. Based on data on ethnicity from the S4 survey, the great majority of the study participants were of German ancestry (Caucasians), with some proportion (~14%) of migration, mainly from the eastern European region. The four cross-sectional surveys included 18,000 participants and serve as cohorts for long-term follow-up studies and as a pool for nested case-control and case-cohort studies. Follow-up activities include address inquiries for all participants (including the assessment of vital status and cause of death), postal questionnaires focusing on chronic diseases, and complete follow-up studies with interviews and physical examinations. Detailed information about KORA has been published previously [26]. The current study was designed as a case-control study nested in the prospective KORA cohort, in which men and women diagnosed with incident first primary CRC and controls were recruited from the S2 (recruitment period: 1989/90), S3 (recruitment period: 1994/1995), S4 (recruitment period: 1999–2001), F3 (follow-up of S3, conducted in 2004/2005) and F4 (follow-up of S4, conducted in 2006–2008) study.

Data on sociodemographic characteristics and medical histories were assessed by standardized computer-aided interviews. The study participants provided information on physical activity during leisure time, alcohol consumption, and smoking status. The leisure-time activity was collected separately for summer and winter and was combined and categorized as active and inactive.

Approval for the KORA study was obtained from the Ethics Committee of the Bavarian Medical Association and the Bavarian commissioner for data protection and privacy. All the participants provided written informed consent.

### 2.2. Cancer Case Ascertainment and Selection

Incident cancer cases in the KORA were identified by the active follow-up of study subjects and verified with cancer registry data. Our outcome of interest was incident first primary CRC (adenocarcinomas), irrespective of molecular subtype or stage, and was defined according to International Classification of Diseases, ICD-9 (153, 154) or ICD-10 (C18-C20). A total of 77 cases (adenocarcinomas, all stages) were individually matched to 77 cancer-free (except non-melanoma skin cancer), living controls by participation in the same study (S2–S4, F3, and F4), age at blood collection, and sex.

### 2.3. Laboratory Measurement of Circulating Bilirubin Concentrations

Circulating UCB levels were measured in serum/plasma samples following a well-established protocol [12,27] using high-performance liquid chromatography (HPLC, Merck, Hitachi, LaChrom, Vienna, Austria), with a Fortis C18 HPLC column (4.6 × 150 mm, 3 μm), Phenomenex SecurityGuard™ cartridges for the C18 HPLC columns (4 × 3 mm), and a photodiode array detector (PDA, Shimadzu). An isocratic mobile phase contained glacial acetic acid (6.01 g/L) and 0.1M n-dioctylamine in HPLC-grade methanol/water (96.5/3.5%). Before starting the procedure, all aliquots were centrifuged and a 50 μL sample was mixed with 200 μL of mobile phase. After a second centrifugation, 120 μL of the supernatant was injected into the HPLC at a flow rate of 1 mL/min.

Case-control pairs were analyzed in the same plate to minimize batch-to-batch fluctuation. Bilirubin (alpha) (purity≥ 98%, Sigma Aldrich) acted as an external standard (3.3% IIIα, 92.8% Ixα, and 3.9% XIIIα isomers, 450nm). One reference serum sample was assessed per analysis as an internal standard. The coefficient of variation (CV) between each plate was 3.76%.

### 2.4. Statistical Analyses

We decided a priori to perform all statistical analyses separately in men and women because of the well-established sex-differences in the levels of UCB, with average higher values in men [28,29] and previous evidence of such differences with regard to cancer risk [25].

The baseline characteristics are presented as means and standard deviations for continuous variables or percentages (%) for categorical variables. Analyses were conducted using unconditional logistic regression models to estimate odds ratios (OR) and 95% confidence intervals (CI) for associations between log-transformed UCB levels (log-UCB), standardized per one-standard-deviation (one-SD) increment, and CRC risk with age, sex, and study adjustment in crude models and further adjustment for body mass index (BMI), height, alcohol consumption, physical activity (active and inactive), smoking status (current, former, and never), and dietary patterns (healthy and unhealthy) (individual components were not available), as well as hormone therapy (HT, yes/no) and menopausal status (pre- and post-menopausal) in women in fully adjusted models. Potential effect modification by sex and menopausal status (among women) was tested by adding a multiplicative interaction term to the fully adjusted model.

All statistical analyses and plots were performed using Stata SE14 (Stata Corporation, College Station, TX, USA). The significance testing was based on two-sided P-values of less than 0.05.

## 3. Results

We identified 77 participants who developed CRC during the follow-up period (between 7 (S4 and F4) and 26 (S2) years). Table 1 shows the main characteristics of the KORA study population. The participants were middle-aged and mostly free of chronic conditions. The mean age was 59.2 years (SD, 11.7 years) in male cases and 58.5 years (SD, 10.5 years) in female cases. Among the men, the cases compared to controls had marginally lower UCB concentrations and were less likely to have never smoked. Among the women, the cases compared to controls had marginally lower UCB concentrations and were more likely to have an unhealthy eating pattern.

Among the men, each one-SD increment in log-UCB was suggestively inversely associated with CRC risk in the crude model (OR, 0.81; 95% CI, 0.47–1.40) as suggested by the lower UCB concentrations among the cases in the unadjusted analysis (Table 1). However, in the fully adjusted unconditional logistic regression model, the association was reversed (adjusted OR, 1.20; 95% CI, 0.52–2.79). Both estimates showed wide confidence intervals consistent with a null association. Among the women, suggestive inverse associations were observed in both crude (OR, 0.86; 95% CI, 0.48–1.56) and fully adjusted models (adjusted OR, 0.76; 95% CI, 0.34–1.84) (Table 2). None of the associations among the men and women reached formal statistical significance (all P≥ 0.4). The P_heterogeneity_ values for differences by sex and menopausal status (among women) were equal to 0.4 and 0.7, respectively.

## 4. Discussion

We investigated the potential etiological role of circulating UCB concentrations in CRC development using serum/plasma samples collected prior to cancer onset (diagnosis) among participants of the KORA cohort. In this population-based prospective cohort study among middle-aged men and women from southern Germany, we found little evidence for associations between circulating UCB concentrations and CRC risk. The confidence intervals around the point estimates were wide due to the limited number of cases (N = 77), and the findings did not reach formal statistical significance. However, the results of this study can be used in future meta-analyses on circulating bilirubin and the risk of CRC.

The observed results of the fully adjusted models (non-significantly positive in men and non-significantly inverse in women) are congruent, in terms of the directions of associations, with our previously published work, where we also used a case-control study design, nested in the prospective EPIC cohort with 1386 cases and 1386 matched controls [25]. In this previous work [25], we observed a positive association among men (OR, 1.19; 95% CI, 1.04–1.36) and an inverse association among women (OR, 0.86; 95% CI, 0.76–0.97). The few other studies to date that have investigated the association between circulating bilirubin concentrations and CRC risk have reported inconsistent results [20,21,31]. In a retrospective case-control study (N cases = 174), each one-unit (μmol/L) decrease in serum bilirubin levels was associated with a 7% increase in CRC risk in men and women [20]. A prospective study in the National Health and Nutrition Examination Survey (NHANES I, N cases = 110) found no association between total bilirubin levels and the incidence of CRC [21], whereas a prior cross-sectional analysis in the NHANES III (N cases = 83) found that every one-unit (mg/dL~ 17.1 μmol/L) increase in serum bilirubin levels was associated with a 75% lower risk of a history of previous CRC [31].

Despite the fact that in vitro work suggested that bilirubin might have antioxidant and anti-inflammatory effects, these inconsistencies in epidemiological studies are most likely attributable to differences in study design, sample size, or bilirubin behavior in human bodies. We used a prospective design with pre-diagnostic blood samples in the current analysis; however, the null results of the current study could be due to the lack of statistical power.

Bilirubin is more than just the final product of heme catabolism. Today, it is considered to be a fundamental endogenously formed product in the blood, with antioxidant and anti-inflammatory properties [32]. UCB has been shown (in vivo) to readily traverse cell membranes, enter into colon cancer cells and inhibit tumor cell proliferation [33], and induce apoptosis in cancer cells (in vitro) [34] and is able to regulate gene transcription (via ERK, p53, and p27) [33,35]. This could explain the serological findings in women if genuine. Additionally, women may be less vulnerable to oxidative stress (due to estrogen, less NADPH-oxidase activity, or other unknown mechanisms) [36]. However, the biology of potentially positive associations in men is currently unknown, and further studies are warranted to elucidate the underlying mechanisms. A possible biological explanation for this unexpected finding could be that the observation is driven by men with high–normal bilirubin levels, which could trigger pro-oxidative processes at high–normal levels in the gut, similar to what has been described for ascorbic acid [37]. Due to the low sample size, we were not able to investigate this further. However, in our earlier work, we could observe that the positive association among men was restricted to men who were homozygous for bilirubin-increasing alleles [25]. Since bilirubin levels are frequently tested in primary healthcare to assess liver function [38], they could also serve as a low-cost biomarker for CRC risk stratification. In the era of precision prevention, the aim of cancer screening is to more precisely assess benefits (e.g., mortality reduction) and risks (e.g., overtreatment) by accounting for patients’ characteristics [39]. Risk stratification for cancer screening can then be more accurately targeted to those whose cancer risk is high enough that the benefits outweigh potential harms [39]. Nonetheless, further studies are required to establish whether bilirubin can serve as a biomarker for more individualized cancer risk stratification.

The strengths of our study include the case-control study design nested within a population-based cohort, the sizable follow-up period, and the direct measurements of serum UCB concentrations. We also had data for a number of demographic, lifestyle, and cancer-related variables to control for confounding. Our study was limited by, first, the low number of CRC cases, which reduced the statistical power of our analysis. Second, the storage of samples for prolonged periods of time (although stored at −190 °C in liquid nitrogen until analysis) could have contributed to a degradation of UCB concentrations, which, however, should have affected the UCB concentrations of cases and controls equally.

## 5. Conclusions

We found little evidence for associations of circulating bilirubin with CRC risk. However, the observed associations (non-significantly positive in men and non-significantly inverse in women) are congruent, in terms of direction, with previous evidence and may be used in future meta-analyses. Given the limited sample size in our study, we cannot exclude the possibility that bilirubin may play a role in CRC development, and further studies are required.

## Figures and Tables

**Table 1 antioxidants-09-00908-t001:** Baseline characteristics of colorectal cancer cases and their individually matched controls by sex in the Cooperative Health Research in the Region of Augsburg (KORA) nested case-control study.

Parameters	Men			Women		
	Case	Control	P	Case	Control	P
N	49	49		28	28	
Age at blood collection (years)	59.2 (11.7)	59.4 (11.8)	>0.9	58.5 (10.5)	58.6 (10.6)	>0.9
UCB (µmol/L)	5.0 (3.5)	5.7 (4.6)	0.3	3.2 (2.2)	3.9 (3.3)	0.5
Weight (kg)	85.0 (12.9)	82.2 (11.7)	0.3	71.0 (13.2)	72.0 (13.3)	0.8
Height (cm)	171.7 (7.5)	173.0 (6.1)	0.4	159.7 (5.9)	160.3 (7.4)	0.8
BMI (kg/m²)	28.8 (3.7)	27.5 (3.4)	0.1	27.8 (4.8)	28.0 (4.6)	0.8
Alcohol intake (g/day)	34.3 (29.1)	34.6 (28.5)	>0.9	11.1 (13.6)	13.9 (16.9)	0.7
Smoking status (n, %)			0.02			0.3
Current	10 (26)	6 (15)		6 (22)	2 (8)	
Former	22 (56)	14 (36)		4 (15)	6 (23)	
Never	7 (18)	19 (49)		17 (63)	18 (69)	
Physical activity (n, %) †			0.2			0.2
Active	13 (34)	19 (49)		12 (44)	7 (27)	
Inactive	25 (66)	20 (51)		15 (56)	19 (73)	
Menopause stage (n, %)						0.6
Post-menopausal				20 (83)	19 (79)	
Pre-menopausal				4 (17)	4 (17)	
Use of HT (n, %)						>0.9
Yes				3 (11)	3 (11)	
No				25 (89)	25 (89)	
Healthy eating patterns (n, %) ‡			0.5			0.04
Healthy	21 (64)	23 (72)		13 (62)	18 (90)	
Unhealthy	12 (36)	9 (28)		8 (38)	2 (10)	

Values are means (SD) unless stated otherwise. Abbreviations: N, n: number; UCB: unconjugated bilirubin; BMI: body mass index; g: gram; HT: hormone therapy. Categorical variables are expressed as n (%), and continuous variables, as means (SD) or medians (5, 95%). Paired t-tests (mean comparison) and chi-square tests for categorical variables were used to calculate the P-values. Numbers of missing values (cases/controls): physical activity (12/12), smoking status (11/12), menopause stage (6/6), healthy eating patterns (23/25). Missing values were not excluded in percentage calculations; therefore, the percentage sum across subgroups is not 100%. † Active: regular physical activity (≥1 h/week); inactive: irregular physical activity (< 1 h/week); no or very little physical activity. ‡ Based on food frequency questionnaire in the KORA surveys according to the recommendations of the German Nutrition Society (DGE), the optimal frequency of 16 food groups was considered as “optimal” (right frequency) = 2 points, “normal” (middle frequency) = 1 point, and adverse (wrong frequency) = 0 points. If the points were ≤13, the unhealthy eating category was chosen, and if the points were ≥14, the normal and healthy eating category was chosen. The score was validated at the individual and population level in two KORA surveys by using additional 7-day food diaries [30].

**Table 2 antioxidants-09-00908-t002:** Odds ratios and 95% confidence intervals for the association between bilirubin levels and colorectal cancer (CRC) risk in the KORA study.

		OR (95% CI)			
	N Cases/ Controls	Crude	P	Adjusted	P
**Total**	77/77	0.83 (0.53–1.31)	0.4	0.97 (0.50–1.86)	0.9
**Men**	49/49	0.81 (0.47–1.40)	0.4	1.20 (0.52–2.79)	0.7
**Women**	28/28	0.86 (0.48–1.56)	0.6	0.76 (0.34–1.84)	0.5

Analyses were conducted using unconditional logistic regression models to estimate odds ratios (OR) and 95% confidence intervals (CI) for associations between log-transformed UCB levels (log-UCB), standardized per one-standard-deviation (one-SD) increment, and CRC risk with age, sex, and study adjustment in the crude model. Full models were further adjusted for BMI, height, alcohol consumption, physical activity, smoking status, and dietary patterns (individual components were not available), as well as hormone therapy (HT) and menopausal status in women. P_heterogeneity_ for sex differences was equal to 0.4.

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
