# Peer review of "The Association between Serum Bilirubin Levels and Colorectal Cancer Risk: Results from the Prospective Cooperative Health Research in the Region of Augsburg (KORA) Study in Germany"

_antioxidants, 2020, doi:10.3390/antiox9100908_

Round 1
Reviewer 1 Report
The manuscript entitled :" The Association between Serum Bilirubin Levels and
Colorectal Cancer Risk: Results from the Prospective Cooperative Health Research in the Region of Augsburg (KORA) Study " focused on the evaluation of UCB in the prediction of CRC cancer risk " requires several modifications to be suitable for pubblication:
- In the manuscript the authors investigate the clinical role of UCB in prediction of CRC development risk. In my opinion, several criticism should be better discussed. In the selected patient cohort, the authors ahve analyzed 77 patients that developed CRC. In my opinion ,the authors should better propose a validation patient cohort in order to clarify the basal levels of UCB in healty donor patients.
- In relation to the last point, in my opinion the authors should clarify sensitivity and specificity of obtained results in relation to other no neoplastic pathologic process. For this purpose, other technical analysis should be performed on a wide patient cohort.
- The authors introduce the concept of clinical application of this analysis on CRC patients. In order to justify the clinical relevance of this analysis, please, could the authors better report some integrative clinical informations about enrolled patients ( histological classification, TNM, stage) About the last critical point, could the authors underline if UBC may be adopted to evaluate CRCdevelopment starting fro mearly stage?
- Could the authors betetr evaluate if molecular assessment related to MSI status or KRAS, NRAS, BRAF in CRC patients at stage IIIB-IV may influence UBC analysis? This aspect is relevant in the statistical analysis to define UBC role in the clinical setting.
Author Response
We thank the Reviewers for the positive evaluation of our manuscript. We also appreciate the constructive and thoughtful comments, which we happily addressed in the revised manuscript as detailed below. The respective changes in the manuscript are in the ‘track changes’ mode.
Reviewer #1, the manuscript entitled: "The Association between Serum Bilirubin Levels and Colorectal Cancer Risk: Results from the Prospective Cooperative Health Research in the Region of Augsburg (KORA) Study" focused on the evaluation of UCB in the prediction of CRC cancer risk requires several modifications to be suitable for publication.
In the manuscript the authors investigate the clinical role of UCB in prediction of CRC development risk. In my opinion, several criticism should be better discussed. In the selected patient cohort, the authors have analyzed 77 patients that developed CRC. In my opinion, the authors should better propose a validation patient cohort in order to clarify the basal levels of UCB in healthy donor patients.
RESPONSE: We wish to clarify that the goal of our data analysis was to investigate potential etiological associations between pre-diagnostic circulating levels of UCB and CRC risk. Thus, the UCB levels were measured among healthy controls and among CRC cases prior to cancer onset (Table 1).
We did not aim to develop a risk prediction model for CRC development, which would have required a different approach – as correctly pointed out by the Reviewer. We clarified this important distinction throughout the manuscript including the Abstract (p. 1, lines 17-19), Introduction (p. 2, lines 59-64), and Discussion (p. 5, lines 169-171).
2- In relation to the last point, in my opinion the authors should clarify sensitivity and specificity of obtained results in relation to other no neoplastic pathologic process. For this purpose, other technical analysis should be performed on a wide patient cohort.
RESPONSE: Given that the goal of the manuscript was to investigate the etiological role of UCB in CRC development, sensitivity and specificity analyses are usually not required. We also would like to point out that only the first primary CRC cases were included in the analysis and that these cases were individually matched to cancer-free controls. We added this information to the Material and Methods (p. 2, line 92). Please also refer to our response to comment #1.
3- The authors introduce the concept of clinical application of this analysis on CRC patients. In order to justify the clinical relevance of this analysis, please, could the authors better report some integrative clinical information about enrolled patients (histological classification, TNM, stage) About the last critical point, could the authors underline if UBC may be adopted to evaluate CRC development starting from early stage?
RESPONSE: We added available information on the classification to the Material and Methods (p. 2, lines 92). Data on stage were no available, which however does not invalidate our results, because our outcome of interest was incident CRC independent of the stage at diagnosis and UCB was measured before onset (or at least before a diagnosis of CRC). Thus, UCB could have an etiological role in CRC development starting from an early stage. We clarified this in the Abstract (p. 1, lines 17-19), Introduction (p. 2, lines 59-64), and Discussion (p. 5, lines 169-171).
4- Could the authors better evaluate if molecular assessment related to MSI status or KRAS, NRAS, BRAF in CRC patients at stage IIIB-IV may influence UBC analysis? This aspect is relevant in the statistical analysis to define UBC role in the clinical setting.
RESPONSE: We agree with the Reviewer that molecular subtype and stage of CRC are very important for the prognosis and treatment of cancer patients. However, these data are less relevant in our analysis, because our outcome of interest was first primary CRC, irrespective of molecular subtype or stage. We clarified this in the Material and Methods (p. 2, lines 89-91).
Reviewer 2 Report
This paper describes the results from an investigation into the potential link between unconjugated bilirubin(UCB) and CRC risk using a case-control approach within a cohort study.The approach is clearly stated and appropriate. It would perhaps be useful to the reader to include in the introduction a brief explanation as to how UCB can act as an anti-oxidant and anti-inflammatory agent. A few minor points to address are as follows:
Line 52-61. Please clarify how the cohort used for the in-press article-reference 20 and the cohort for this study relate to each other if they do. IF they do not, please make this much more explicit.
Explain at an earlier point that KORA is based in Germany, perhaps on line 61.
The p values given in table 2 are not significant and really show no effect. The discussion suggesting a trend is misleading and should be modified. Line 164 is written to suggest they were close to being significant which would imply a value of just a little great than p=0.05. It is however important, as the authors state, to publish this data for use in future meta analyses.
There is some discussion on pre and post menopausal women. If the few post menopausal women are removed from the analysis does this have any impact on the results?
Author Response
We thank the Reviewers for the positive evaluation of our manuscript. We also appreciate the constructive and thoughtful comments, which we happily addressed in the revised manuscript as detailed below. The respective changes in the manuscript are in the ‘track changes’ mode.
Reviewer #2, This paper describes the results from an investigation into the potential link between unconjugated bilirubin (UCB) and CRC risk using a case-control approach within a cohort study. The approach is clearly stated and appropriate. It would perhaps be useful to the reader to include in the introduction a brief explanation as to how UCB can act as an anti-oxidant and anti-inflammatory agent. A few minor points to address are as follows:
1-Line 52-61. Please clarify how the cohort used for the in-press article-reference 20 and the cohort for this study relate to each other if they do. IF they do not, please make this much more explicit.
RESPONSE: We clarified that the cohort for this study (KORA) and the cohort (EPIC) for the in press article (ref. 20) are independent and there is no overlap of participants (p. 2, lines 59-61) (p. 5, lines 178-184).
2-Explain at an earlier point that KORA is based in Germany, perhaps on line 61.
RESPONSE: In agreement with the Referee’s comment, we added this statement in the Title (p.1, line 5), Abstract (p. 1, line 20), Introduction (p. 2, line 64), and Discussion (p. 5, lines 172-173).
3-The p values given in table 2 are not significant and really show no effect. The discussion suggesting a trend is misleading and should be modified. Line 164 is written to suggest they were close to being significant which would imply a value of just a little great than p=0.05. It is however important, as the authors state, to publish this data for use in future meta analyses.
RESPONSE: In agreement with the Referee’s comment, we reworded the description of Results (p. 3, lines 133-140), Discussion (p. 5, lines 169-174; lines 178-180; lines 205-207), and Conclusion using an appropriately cautious interpretation (p. 6, lines 220-224).
4-There is some discussion on pre and post-menopausal women. If the few post-menopausal women are removed from the analysis does this have any impact on the results?
RESPONSE: We checked for potential effect modification of results by menopausal status among women and re-run the fully adjusted model after excluding post-menopausal women. We observed no change in associations (P-heterogeneity = 0.72). We added this test for multiplicative interaction to the Material and Methods (p. 3, lines 120-122) and report the P-heterogeneity in the Results (p. 3-4, lines 140-142).
Reviewer 3 Report
This study by Khoei et al. aims to investigate the association between serum bilirubin Levels and CRC. The study is well designated but is not extensively analyzed. The authors should provide more information in the Introduction section and sufficiently state their conclusions, in order to be used in a meta-analysis.
Author Response
We thank the Reviewer for the positive evaluation of our manuscript. We also appreciate the constructive and thoughtful comment, which we happily addressed in the revised manuscript as detailed below. The respective changes in the manuscript are in the ‘track changes’ mode.
Comments and Suggestions for Authors
This study by Seyed Khoei et al. aims to investigate the association between serum bilirubin Levels and CRC. The study is well designated but is not extensively analyzed.
- The authors should provide more information in the Introduction section and sufficiently state their conclusions, in order to be used in a meta-analysis
RESPONSE
We thank the Reviewer for the encouraging feedback. Following the Reviewer’s recommendation, we revised the Introduction and provide a more detailed rationale for studying the relationship between circulating bilirubin and the risk of colorectal cancer (pages 1-2, lines 41-47). We also revised the conclusions as suggested (page 7, lines 230-232).
Reviewer 4 Report
The authors have put together a manuscript on bilirubin's importance in colorectal cancer in men and women. This is a topic that falls within the scope of the journal. There was another manuscript by the senior author (Karl-Heinz Wagner) on a similar topic. The authors make an interesting conclusion that bilirubin in this study has a positive association with CRC risk among men and an inverse association for women. Overall, the paper is well-written, and the study is well-designed. This reviewer only has minor comments: 1) the UCB levels in the study are lower in both men (5.7 for control versus 5.0 for cases) and women (3.9 for control versus 3.2 for cases) with CRC as described by 'case' compared to 'control.' The authors should state that bilirubin is lower in CRC cases, but after adjustments, they found differences. Also, the bilirubin levels overall are very low; it would be good to discuss this fact as well. If the authors have the data for the patients' ethnicity, this would be valuable to add. These comments could improve the manuscript and are minor comments. The manuscript is a well-written document and an interesting study.
Author Response
We thank the Reviewer for the positive evaluation of our manuscript. We also appreciate the constructive and thoughtful comments, which we happily addressed in the revised manuscript as detailed below. The respective changes in the manuscript are in the ‘track changes’ mode.
The authors have put together a manuscript on bilirubin's importance in colorectal cancer in men and women. This is a topic that falls within the scope of the journal. There was another manuscript by the senior author (Karl-Heinz Wagner) on a similar topic. The authors make an interesting conclusion that bilirubin in this study has a positive association with CRC risk among men and an inverse association for women. Overall, the paper is well-written, and the study is well-designed. This reviewer only has minor comments:
We thank the Reviewer for the encouraging feedback.
- The UCB levels in the study are lower in both men (5.7 for control versus 5.0 for cases) and women (3.9 for control versus 3.2 for cases) with CRC as described by 'case' compared to 'control.' The authors should state that bilirubin is lower in CRC cases, but after adjustments, they found differences.
RESPONSE
We agree with the Reviewer’s comment and revised the text accordingly (page 3, lines 139-140).
- Also, the bilirubin levels overall are very low; it would be good to discuss this fact as well.
RESPONSE
In agreement with the Reviewer’s comment, we added this part to our study limitations in the Discussion (page 5, lines 225-228).
- If the authors have the data for the patients' ethnicity, this would be valuable to add. These comments could improve the manuscript and are minor comments. The manuscript is a well-written document and an interesting study.
RESPONSE
Based on data on ethnicity from the S4 survey, the great majority of our participants were of German ancestry (Caucasians) with some proportion (~14%) of migration mainly from the eastern European region. We added this information to the Material and Methods part (page 2, lines 74-76).
Round 2
Reviewer 1 Report
The manuscript entitled "The Association between Serum Bilirubin Levels and
Colorectal Cancer Risk: Results from the Prospective Cooperative Health Research in the Region of Augsburg (KORA) Study in South Germany" fcoused on the relation between serum bilirubin levels and colorectal cancer risk is not suitable for pubblication. In my opinion, several critical points should be implemented. The study is lack of a technial part where this correlation was well documented. In addition, reported data are not clearly discussed to confirm author's hypothesis. Finally, clinical relevance of this manuscript is not adeguately discussed beacause very few informations are debaeted.
Author Response
We thank the Reviewer for the positive evaluation of our manuscript. We also appreciate the constructive and thoughtful comments, which we happily addressed in the revised manuscript as detailed below. The respective changes in the manuscript are in the ‘track changes’ mode.
Comments and Suggestions for Authors
The manuscript entitled "The Association between Serum Bilirubin Levels and
Colorectal Cancer Risk: Results from the Prospective Cooperative Health Research in the Region of Augsburg (KORA) Study in South Germany" fcoused on the relation between serum bilirubin levels and colorectal cancer risk is not suitable for pubblication. In my opinion, several critical points should be implemented.
- The study is lack of a technial part where this correlation was well documented.
RESPONSE
We thank the Reviewer for engaging with our study. In response to what we think was the Reviewer’s suggestion, we provided a more detailed and technical rationale of why we investigated the relationship between bilirubin and colorectal cancer (pages 1-2, lines 41-47).
- In addition, reported data are not clearly discussed to confirm author's hypothesis.
RESPONSE
In response to the Reviewer’s suggestion, we expanded our discussion with regard to our hypothesis (pages 5-6, lines 209-212).
- Finally, clinical relevance of this manuscript is not adeguately discussed beacause very few informations are debaeted.
RESPONSE
In response to the Reviewer’s recommendation, we added a short paragraph about the clinical relevance (page 6, lines 212-221).

Reviewer 3 Report
The issues raised by the Reviewers after the previous submission of this manuscript were properly addressed. The manuscript has been improved and is now suitable for publication in “Antioxidants”.
Round 3
Reviewer 1 Report
The manuscript entitled:" The Association between Serum Bilirubin Levels and Colorectal Cancer Risk: Results from the Prospective Cooperative Health Research in the Region of Augsburg (KORA) Study in Germany is now suitable for publication without any revision.Theauthors have clarified each revision point in the text.